# Fabrication of Well-Aligned ZnO Nanorods with Different Reaction Times by Chemical Bath Deposition Method Applying for Photocatalysis Application

**DOI:** 10.3390/molecules28010397

**Published:** 2023-01-03

**Authors:** Htet Su Wai, Chaoyang Li

**Affiliations:** 1School of Systems Engineering, Kochi University of Technology, 185 Miyanokuchi, Tosayamada cho, Kami City 782-8502, Kochi, Japan; 2Center of Nanotechnology, Kochi University of Technology, 185 Miyanokuchi, Tosayamada cho, Kami City 782-8502, Kochi, Japan

**Keywords:** ZnO nanorods, AZO seed layer, growth alignment, chemical bath deposition, photocatalytic efficiency

## Abstract

Zinc oxide nanorods were grown on an aluminum-doped zinc oxide seeds layer using the chemical bath deposition method. The effects of growth reaction time on the structural, optical, and photocatalytic properties of zinc oxide nanorods were investigated. It was clearly observed that the growth direction of zinc oxide nanorods were dependent on the crystallinity of the as-deposited aluminum-doped zinc oxide seed layer. The crystallinity of the obtained zinc oxide nanorods was improved with the increase in reaction times during the chemical bath deposition process. The mechanism of zinc oxide nanorod growth revealed that the growth rate of nanorods was influenced by the reaction times. With increasing reaction times, there were much more formed zinc oxide crystalline stacked growth along the c-axis orientation resulting in an increase in the length of nanorods. The longest nanorods and the high crystallinity were obtained from the zinc oxide nanorods grown within 5 h. The optical transmittance of all zinc oxide nanorods was greater than 70% in the visible region. Zinc oxide nanorods grown for 5 h showed the highest degradation efficiency of methyl red under ultraviolet light and had a high first-order degradation rate of 0.0051 min^−1^. The photocatalytic mechanism was revealed as well.

## 1. Introduction

Since the last decade, there has been much interest in using photocatalysis to solve environmental concerns related to removing hazardous wastewater compounds and toxic organic dyes [1,2,3,4]. Among the various dyes, methyl red (MR) is one of the Azo dyes, which have been widely used in water purification, carbon dioxide reducing, textiles, and some food products [5,6]. However, MR had been classified as harmful and toxic dyes because it causes eye irritation and harm to the skin and digestive tract if swallowed or inhaled [7,8]. Therefore, it will be very helpful to effectively discover the MR degradation method using a metal oxides semiconductor to prevent the environmental pollutant problems. Among the metal oxides, mixed oxides, and hybrid non/metal-oxides, semiconductor-based metal oxides, such as zinc oxide (ZnO), titanium dioxide, iron oxide, tungsten oxide, and cadmium sulfide, have great potential for degrading chemical pollutants and microorganism decontamination due to non-toxicity, high-oxidization efficiency, and high photon sensitivity [9,10,11,12,13,14,15]. Particularly, ZnO has been considered a promising semiconductor-based metal oxide due to its unique properties, such as a wide bandgap (3.37 eV) at room temperature, large exciton binding energy (60 meV), high surface reactivity, super hydrophilicity, environmentally friendly features, and low cost [16,17,18]. Recently, ZnO-based nanostructures have attracted much attention as photocatalysts for removing organic dye pollutants from water [19,20,21]. Moreover, numerous reports have already demonstrated that the photocatalytic efficiency of ZnO was strongly influenced by the surface morphology and various ZnO nanostructure modifications (spheres, rods, tubes, needles, etc.) in improving the performance of photocatalytic application [22,23,24]. Among the different nanostructures of ZnO, one-dimensional ZnO nanorods generally possess better adsorption of organic dyes and higher photocatalytic activity than the nanoparticles due to a large surface-to-volume ratio [25,26,27,28]. Moreover, the relationship between the exposed surfaces of ZnO nanorods and their photocatalytic efficiency has already been reported [29,30,31]. It was recently reported that the high index (0001) facets possess high surface energy; thus, it has high potential to improve the photocatalytic efficiency. The increase in the proportion of the exposed polar surface leads to the enhancement of photocatalytic efficiency, which might be due to the polar surface of (0001)-Zn being preferred to absorb more OH- ions and thus could be combined with hydroxyl radicals to generate more reactive oxygen species (ROS), which are capable of degrading organic contaminants and damaging microorganisms [32,33,34,35,36]. Li et al. had reported that the modification of surface morphology and controlling the balance between specific surface area and structural defects are mainly involved in the role of photocatalytic efficiency [37]. Moreover, due to their high surface area, well-aligned ZnO rods allow much more light to trap and generate the more charged carriers on the catalyst surface resulting in the increase the ROS species. Pelicano et al. reported that the modified core–shell heterostructure of ZnO nanorods could enhance charge-transporting capabilitywith the light-absorber layer and controlled the pH level to modify the different kinds of ZnO nanostructures for flexible and highly efficient photovoltaic devices [38,39]. Thus, it is a great concern to synthesize the ZnO nanostructures with controllable c-axis growth direction as (0001) facet and a large surface area to enhance the photocatalytic efficiency.

Among the various techniques for fabricating well-aligned ZnO nanorods, chemical bath deposition (CBD) was reported to be the most appealing because it is a simple, low-cost, large-scale production, and low-temperature process [40,41,42]. In our previous research, the growth control of ZnO nanorods was achieved on the selected substrates which had less lattice-matching to ZnO material [43]. Comparing to the ZnO film, the aluminum-doped zinc oxide (AZO) film has much higher thermal and chemical stability. Therefore, AZO film with good crystallinity will be a better seed layer for controlling the growth direction of ZnO nanorods in the CBD method. In this research, we proposed growing the ZnO nanorods on the sputtered AZO seed layer. The influence of the reaction times on the structural and optical properties of ZnO nanorods, as well as the photocatalytic properties of ZnO nanorods, were investigated.

## 2. Experiments

### 2.1. Deposition of AZO Film

An radio frequency (RF) (13.56 MHz) magnetron sputtering system was applied to deposit the 300 nm-thick AZO film on the glass substrate (Alkaline free glass sheet, Eagle XG). A 4-inch AZO (ZnO: Al_2_O_3_ = 98: 2 wt%) was used as a target in the sputtering system. Prior to the deposition process, the substrate was pre-heated at 150 °C for one hour. During the deposition process, pure argon (30 sccm) was supplied as a working gas into the chamber, which was adjusted by the mass controller. Pressure, power and temperature were kept constant at 1 Pa, 100W and 150 °C, respectively. The deposition conditions are shown in Table 1.

### 2.2. ZnO Nanorods Growth

Following the sputtering process, the obtained as-deposited AZO film was applied as a seed layer for growing the ZnO nanorods by the CBD method. A mixed precursor solution of Zn(NO_3_)_2_·6H_2_O concentration of 0.015 mol/L and hexamethylenetetramine (HMTA) with a concentration of 0.0075 mol/L was prepared in a solution bottle, and it was dissolved in ultrapure water (200 mL) as solvent. The prepared precursor solution was poured into a flask and kept at 95 °C. In order to investigate the effect of reaction time on the growth of ZnO nanorods, the reaction time was set from 1 h to 5 h with one-hour interval.

### 2.3. Photodegradation Process

The photocatalytic activity of ZnO nanorods was evaluated in a glass beaker by the degradation of a methyl red (MR) solution at room temperature. The concentration of a precursor solution of 1 × 10^−5^ mol/L was prepared by dissolution in water (70 mL). The bottle of mixed precursor solution was wrapped with aluminum foil to eliminate light for 30 min and stirred it before irradiation. After the stirring process, the obtained ZnO nanorods with different reaction times were put inside the precursor solution individually, then it was irradiated under ultraviolet light with a wavelength of 254 nm for 5 h. In order to investigate the absorption of MR dye on the ZnO nanorods before photocatalysis, each MR solution prepared with the obtained ZnO nanorods was exposed without UV light for 5 h. After the photodegradation process, the irradiated solution was taken out at each one-hour interval to analyze the degradation rate as a function of the reaction time.

### 2.4. Characterization

The thickness of the as-deposited AZO film was measured by spectroscopic ellipsometry (WVASE32, J.A. Woollam, Co., Inc., Lincoln, CA, USA). The morphologies of the as-deposited AZO film and ZnO nanorods were evaluated by field-emission scanning electron microscopy (FE-SEM, SU-8020, Hitachi, performed with 5 kV operating voltage, with magnification from 20 K to 100 K). The structural properties of the as-deposited AZO film and ZnO nanorods were investigated by grazing incidence X-ray diffraction (GIXRD, ATX-G, Rigaku, performed in 2θ/ω sweeping from 20 to 40 degree in 2θ with a scan speed at 0.5°/min, employing a CuKα tube (λ = 0.154178 nm) radiation (50 kV, 300 mA)). A UV–visible spectrophotometer (U-4100, Hitachi) was used to evaluate the optical properties of the as-deposited AZO film and ZnO nanorods, as well as the absorption spectra of the MR solution. All of the measurements were carried out at room temperature.

## 3. Results and Discussion

The XRD patterns of the as-deposited AZO film and aligned ZnO nanorods grown on AZO seed layer with different reaction times by the CBD method are shown in Figure 1. It can be observed that there was only a (002) diffraction peak for the as-deposited AZO film and all ZnO nanorods with different reaction times, which meant that the growth direction of the obtained ZnO nanorods followed the same c-axis growth tendency as the below as-deposited AZO film. All of the ZnO nanorods preferentially oriented growth in the (0001) crystal plane perpendicular to the substrate. As shown in the Table 2, the intensity of the (002) peak was increased, and that of the FHWM was decreased. According to the Debye Scheer’s formula [44], the calculated c-axis crystallite size of the ZnO nanorods increased from 30.2 nm to 39.5 nm when the reaction time was increased from 1 h to 5 h during the CBD process. The calculated lattice constant c is converged to the relaxed lattice constant of the bulk ZnO (0.52 nm). This result indicated that the crystallinity of the ZnO nanorods was enhanced with the reaction time increase. The highest crystallinity was obtained from the ZnO nanorods grown for 5 h.

Based on the biaxial strain model [45], the compressive stress (σ) in the AZO film and ZnO nanorods grown with different reaction times could be expressed as
(1)$$\sigma =\frac{2C_{13}^{2}-C_{33}\left(C_{11}+C_{12}\right)}{2C_{13}}\;\times \;\frac{C_{film}-C_{bulk}}{C_{bulk}}$$
where c is the lattice constant, Cij is the elastic modulus of the bulk ZnO film, and *C*_11_ = 208.8 GPa, *C*_12_ = 119.7 GPa, *C*_13_ = 104.2 GPa, and *C*_33_ = 213.8 GPa. The lattice constant c could also be calculated based on residential stress and XRD patterns. The peak position of the (002) diffraction peak of the as-deposited AZO film was 34.32°, and it was gradually shifted from 34.36°, 34.39°, 34.41°, and 34.42° to 34.45°, respectively, corresponding to ZnO nanorods with different growth reaction times. According to Equation (1), the calculated compressive stresses were −0.95 GPa, −0.82 GPa, −0.37 GPa, 0.15 GPa, and 0.47 GPa, corresponding to ZnO nanorods grown from 1 h to 5 h. It was indicated that the compressive stress was gradually relieved in ZnO nanorods grown by the CBD process. In addition, the lattice mismatch between the ZnO nanorods and as-deposited AZO film decreased from 0.09% to 0.06% when the reaction time was increased from 1 h to 5 h, which meant that the obtained ZnO nanorods followed the same growth direction as the as-deposited AZO film and that the crystallinity was improved as the reaction time was increased. The low lattice mismatch was achieved from the ZnO nanorods grown on the AZO seed layer for 5 h.

Figure 2 shows the SEM images of the as-deposited AZO film and the aligned ZnO nanorods grown on the AZO seed layer with different reaction times by the CBD method. From the top view of images, it could be observed that the as-deposited AZO film surface was flatly and uniformly, with an average grain size of 52 nm. After the CBD process, as shown in Figure 2b–f, the obtained ZnO nanorods had well-defined hexagonal facets, and the average diameter of the ZnO nanorods slightly increased from 51 nm to 97 nm when the reaction time was increased from 1 h to 5 h. In contrast, the density of the ZnO nanorods was decreased from 310/µm^2^ to 88/µm^2^, corresponding to the reaction times increasing from 1 h to 5 h. The average length of the ZnO nanorods increased significantly from 144 nm to 1300 nm with the increase in reaction time from 1 h to 5 h. Moreover, it was clearly revealed that the obtained ZnO nanorods followed a highly vertical alignment on the as-deposited AZO film as the reaction time increased. The longest ZnO nanorod was obtained from the AZO film grown for 5 h.

The growth mechanism of ZnO nanorods with different reaction times by CBD method was revealed as the following equations. During the CBD process, the mixed precursor solution of Zn(NO_3_)_2_·6H_2_O and HMTA was dissolved in water and continuously generated Zn^2+^ and OH^−^ ions, based on the following equations;
Zn(NO_3_)_2_ → Zn^2+^ + 2 NO_3_^−^(2)
C_6_H_12_N_4_ + H_2_O → 4 NH_3_ + 6HCHO(3)
NH_3_ + H_2_O → NH_4_^+^ + OH^−^
(4)
Zn^2+^ + 2 OH^−^ → Zn(OH)_2_
(5)
Zn(OH)_2_ → ZnO + H_2_O(6)

According to the Equation from (2) to (6), the generated Zn^2+^ ions from the dissolution of Zn(NO_3_)_2·_6H_2_O_,_ were ionized and reacted with the OH^−^ ions, which were hydrolyzed from HMTA, and then formed Zn(OH)_2_. The obtained Zn(OH)_2_ could be easily dehydrated and immediately formed ZnO. When the reaction time was increased, much more ZnO crystallites might be formed and stacked on top of each other, resulting in the increasing of the ZnO nanorods in length.

EDX measurement evaluated the elemental composition of the as-deposited AZO film and ZnO nanorods with different reaction times by CBD method. The variation plot of the atomic ratios of (Zn + Al)/O, Zn/O, and Al/O are shown in Figure 3. It was observed that the atomic ratio of (Zn + Al)/O obviously increased as the reaction time was increased from 1 h to 5 h, which indicated that, more Zn^2+^ ions were released and formed ZnO during the CBD process. Moreover, the atomic ratios of Zn/O were also significantly increased; in contrast, that of Al/O stayed at nearly the same level as the reaction time was increased from 1 h to 5 h, indicating that the influence of Al on the ZnO nanorods was negligible during the CBD process. On the other hand, the atomic ratios of Zn/O were greater than that of Al/O when the reaction time was increased from 1 h to 5 h, which meant that the much more numerous ZnO crystallites were stacked along the c-axis growth direction. The increase in the bonding ratios of Zn/O resulted in the increase in length of ZnO nanorods during the CBD process. The highest atomic ratio of Zn/O was found from the ZnO nanorods grown for 5 h.

Figure 4 shows the optical transmittance spectra of the as-deposited AZO film and ZnO nanorods on the as-deposited AZO film with different reaction times. It could be observed that the high transmittance of the as-deposited AZO film was around 90% and all ZnO nanorods also showed a high transmittance of around 70% in the visible region. It can be seen that the transmittance was less affected by changes in reaction times during the CBD process because the obtained all ZnO nanorods were well-aligned vertically on the sputtered AZO film.

Based on Tauc’s plot equation [46], the calculated bandgap energy (Eg) values of the obtained ZnO nanorods decreased from 3.31 eV to 3.21 eV when the growth reaction time was increased from 1 h to 5 h. As a result, the electrons from the valence band were much more easily transferred to the conduction band of the ZnO nanorods which had the narrower bandgap, resulting in increasing the formation rate of active radicals during the photodegradation process.

### 3.1. Photocatalytic Activity of ZnO Nanorods

Figure 5 shows the absorption spectra of an irradiated MR solution of the ZnO nanorods grown with different reaction times by the CBD method without UV light. The highest absorption band at 520 nm was selected, which corresponds to the red color of the MR solution [47]. It was found that there was negligible degradation of MR for the ZnO nanorods grown for 1 h and 2 h, then the absorption intensities of MR slightly decreased for the ZnO nanorods grown from 3 h to 5 h. This indicated the obtained ZnO nanorods showed the low photocatalytic efficiency without UV light.

Figure 6a presents the absorbance spectra of the MR solution with the ZnO nanorods under UV light irradiation. It can be clearly seen that the absorption intensities of the MR solution in the presence of ZnO nanorods with different reaction times diminished gradually with the increasing of reaction times from 1 h to 5 h comparing to the absorption intensity of the original MR solution. The results indicated that photodegradation efficiency was achieved in all ZnO nanorods. The lowest absorption intensity was achieved from the ZnO nanorods grown for 5 h.

According to the absorption spectra of ZnO nanorods as shown in Figure 6a, it can be clearly found that all the absorption spectra were covered between the wavelength of 300 nm and 700 nm, indicating that the all obtained ZnO nanorods showed photocatalytic efficiency under the both ultraviolet and visible light irradiation.

During the CBD process, the concentration of Zn(NO_3_)_2_·6H_2_O was two times higher than the concentration of HMTA, indicating that the concentration of the generated Zn^2+^ ions was significantly higher than the OH^−^ ions. Thus, the increase in the Zn^2+^ ions might introduce the zinc interstitial (Zn_i_) defects to ZnO nanorods during CBD growth. The Zn_i_ defects perform as an intermediate band in the energy level, where the electron excitation could happen with a lower amount of energy. Hence, the obtained ZnO nanorods with Zn_i_ defects showed improvement of photocatalytic efficiency within the visible range.

Figure 6b illustrates the relationship between the concentration of the MR solution and the function of UV light irradiation time for ZnO nanorods grown with different reaction times. C_0_ represents the starting reactant concentration (mM), and C represents the reactant concentration (mM) at time “t”. Notably, it was obviously seen that the concentration of the MR solution decreased within 5 h when the CBD reaction time was increased from 1 h to 5 h. The lowest concentration of the MR solution after degradation occurred with ZnO nanorods grown for 5 h.

According to the Langmuir–Hinshelwood kinetic model [48], the pseudo-first-order reaction rates of the MR solution with different irradiation time were calculated. The rate of reaction (R) can be expressed as
R = −dC/dt = kr KC(7)
where C is the concentration of the reactant, t is the reaction time, kr is the reaction rate constant, and K is the adsorption coefficient of the reactant. As the integration law, the first-order reaction rate constant (k) was estimated as follows;
ln (C/Co) = kt(8)
k = ln (C/Co)/t(9)

According to the Langmuir–Hinshelwood (L–H) mode, the higher calculated value of the first-order rate corresponds to the higher photocatalytic efficiency. According to Equation (9), the calculated k values of the ZnO nanorods after 5 h of irradiation increased from 0.0021 min^−1^ to 0.0057 min^−1^ when the CBD reaction time was increased from 1 h to 5 h. The highest reaction rate was achieved from the ZnO nanorods grown for 5 h.

### 3.2. Photocatalytic Mechanism of ZnO Nanorod

The photocatalytic mechanism of ZnO is depicted in the schematic diagram shown in Figure 7a. When the UV light is irradiated onto the ZnO, the electrons (e^−^) from the valence band of ZnO are transformed into the conduction band, leaving the holes (h^+^) behind in the valance band. The reaction mechanism for photodegradation within ZnO was described as follows.
(10)$$e^{-}+O_{2}\;\to \;.O_{2}^{-}$$
h^+^ + H_2_O → OH^−^ + H^+^
(11)
(12)$$.O_{2}^{-}+{\mathrm{OH}}^{-}\;\to \;{}^{\circ}{\mathrm{HO}}_{2}\;$$
°HO_2_ + H^+^ → H_2_O_2_
(13)
H_2_O_2_ + e^−^ → OH^−^ + °OH (14)

According to Figure 7a, the photoexcited electron (e^−^) could easily move on the surface of ZnO and oxidized with the O_2_ from the atmosphere to form superoxide radicals (·O_2_^−^). The hole (h^+^) also reacted with water molecules (H_2_O) and produced OH^−^ ions. The generated superoxide radicals (·O_2_^−^) could be continuously reacted with decomposed hydrogen ions from water molecules and form H_2_O_2_, then create the hydroxyl radicals (°OH), as per Equations (10)–(14). In order to improve the photocatalytic efficiency of ZnO, the generation rate of ROS radicals dependent on the active surfaces of ZnO. In the wurtzite structure of ZnO, owing to the (0001) facet whichhas high surface energy and creates the more active surfaces of ZnO with the plentiful active sites, as shown in the Figure 7b. The ZnO polar surface of (0001) can easily absorb OH^−^ ions due to their surface positive charge, then react with the photogenerated hole (h^+^) to generate ROS, which can enhance photocatalytic efficiency. According to the XRD results, the (0001) crystal facet of ZnO nanorods were enhanced with the increasing of growth reaction time. The strongest (0001) crystal facet was obtained from the ZnO nanorods grown for 5 h. On the other hand, in the case of ZnO nanorods with high crystallinity along the (0001) crystal plane, the direct band transition could happen efficiently and increase the generation rate of electron and hole pairs to form ·O_2_− and °OH radicals. In this research, the crystallinities of ZnO nanorods were also improved with the increasing of the growth reaction time from 1 h to 5 h. The obtained well-aligned ZnO nanorods grown for 5 h showed the highest crystallinity and had the strongest (0001) growth direction. This result is attributed to the fact that the ZnO nanorods grown for 5 h could absorb more OH^−^ ions and generate much more hydroxyl radicals to degrade the MR solution.

In addition, the volume, including the length and diameter, of the obtained ZnO nanorods increased when the growth reaction time was increased from 1 h to 5 h, which meant that there were much more active sites on the surface of the ZnO nanorods. The increase in the number of active sites could enhance the surface absorption ability of the MR solution by absorbing more incident photons to generate the photogenerated charge carriers, thus improving the degradation reaction rate. Therefore, the high photocatalytic efficiency and the high first-order were achieved from ZnO nanorods grown for 5 h.

## 4. Conclusions

During the CBD process, the growth reaction time significantly influenced the growth and crystallinity of the well-aligned ZnO nanorods on the as-deposited AZO seed layer. The vertical alignment of the ZnO nanorods followed the same growth direction as the underlying AZO seed layer. The crystallinity of the ZnO nanorods in the (0001) growth orientation was improved when the reaction time was increased from 1 h to 5 h during CBD growth. The highest crystallinity was obtained from the ZnO nanorods grown with 5 h reaction time. When the growth reaction time was increased, more Zn^2+^ ions were generated and strongly bonded with oxygen. The formed ZnO crystallites could be stacked along the c-axis (0001) growth direction, resulting in the length increasing during the CBD process. The length of the ZnO nanorods also increased with the increasing reaction time. All fabricated ZnO nanorods showed a high transmittance of over 70% in the visible region. The obtained ZnO nanorods grown with 5 h reaction time showed the highest photodegradation efficiency and the largest value of a first-order reaction rate of 0.0057 min^−1^. The results indicated that the obtained ZnO nanorods could be efficiently degraded by the methyl red dye solution and had a high potential to be applied in photocatalyst application.

## Figures and Tables

**Figure 1 molecules-28-00397-f001:**
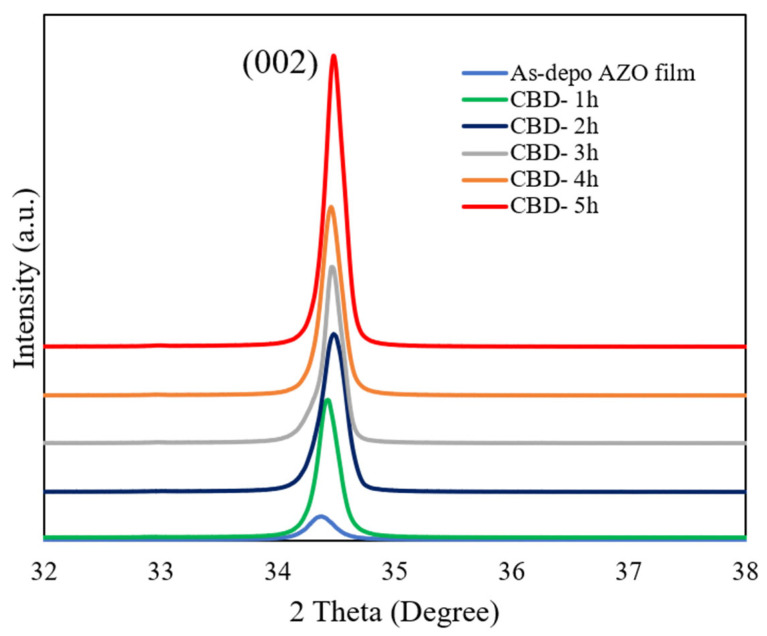
GIXRD patterns of ZnO nanorods with different reaction times by CBD method.

**Figure 2 molecules-28-00397-f002:**
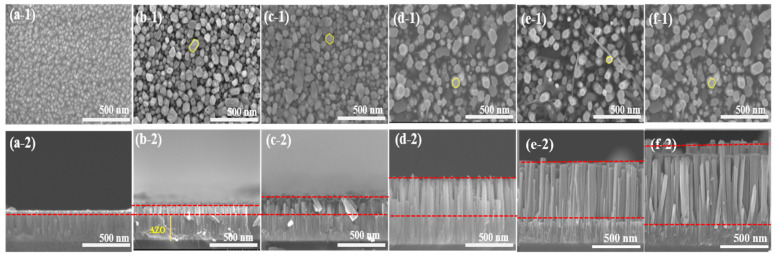
SEM images of ZnO nanorods with diffraction reaction times by CBD method: top views (**a-1**) as-deposited AZO film, (**b-1**) CBD—1 h, (**c-1**) CBD—2 h, (**d-1**) CBD—3 h, (**e-1**) CBD—4 h, and (**f-1**) CBD—5 h; cross-section views: (**a-2**) as-deposited AZO film, (**b-2**) CBD—1 h, (**c-2**) CBD—2 h, (**d-2**) CBD—3 h, (**e-2**) CBD—4 h, and (**f-2**) CBD—5 h.

**Figure 3 molecules-28-00397-f003:**
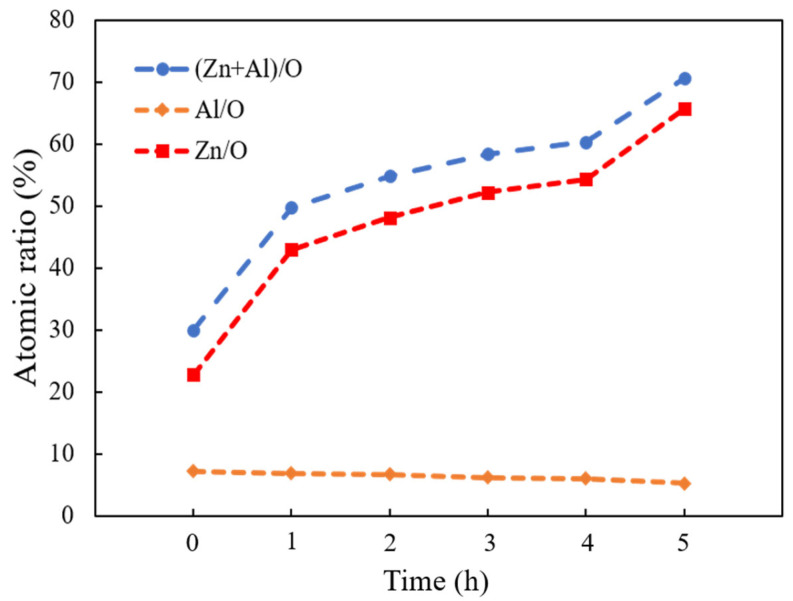
Variations of (Zn + Al)/O, Zn/O, and Al/O atomic ratios calculated from EDX analysis of as-deposited AZO film and ZnO nanorods with different reaction times by CBD method.

**Figure 4 molecules-28-00397-f004:**
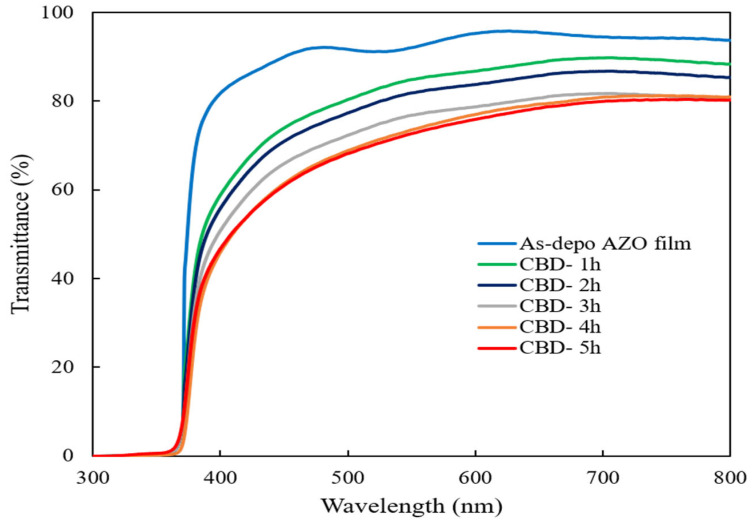
Optical transmittance spectra of ZnO nanorods with different reaction times by CBD method.

**Figure 5 molecules-28-00397-f005:**
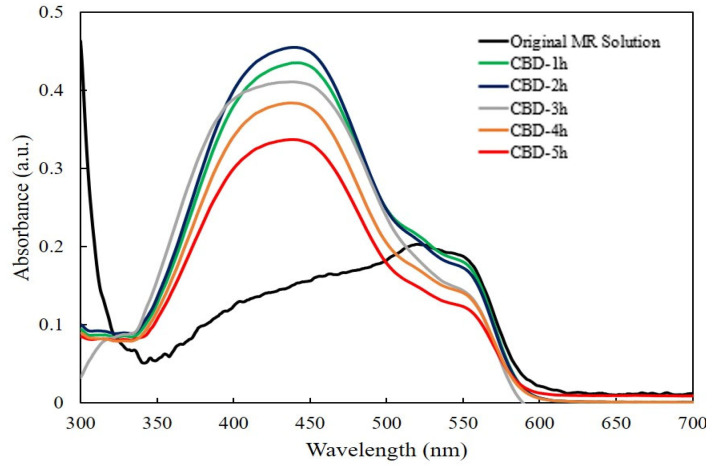
The absorption spectra of irradiated MR solution for ZnO nanorods with different reaction times without UV light.

**Figure 6 molecules-28-00397-f006:**
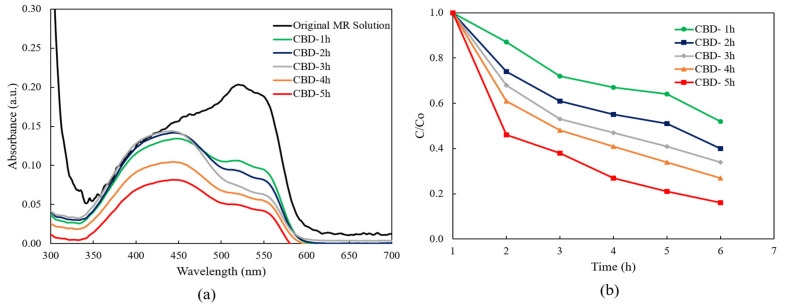
(**a**) The absorption spectra of MR solution for ZnO nanorods with different reaction times and (**b**) plot of C/Co versus the degradation reaction time.

**Figure 7 molecules-28-00397-f007:**
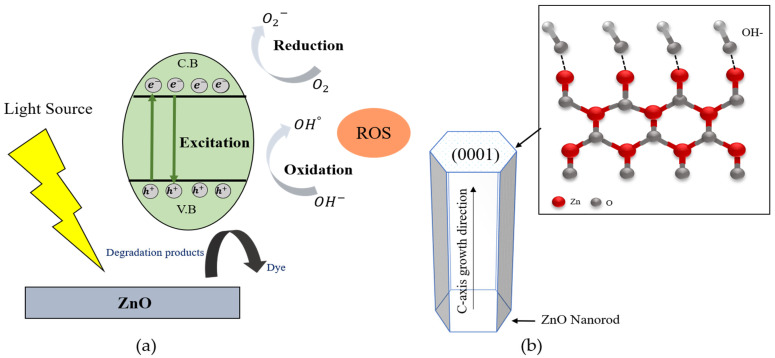
(**a**) Schematic diagram of photocatalytic mechanism of ZnO, and (**b**) (0001) plane of ZnO nanorod for photocatalytic efficiency.

**Table 1 molecules-28-00397-t001:** Deposition conditions of AZO film by RF magnetron sputtering.

Deposition Parameter	Condition
Substrate	Glass
Target	(ZnO:Al_2_O_3_ = 98:2 wt%)
Deposition temperature (°C)	150
Pressure (Pa)	1
Power (W)	100
Working gas, Ar (sccm)	30

**Table 2 molecules-28-00397-t002:** XRD analysis date of as-deposited AZO film and ZnO nanorods.

Samples	Peak Position (Deg)	Intensity (a.u.)	FWHM (Deg)	d (nm)	C-axis Crystallite Size (nm)	Lattice Constant, c (nm)	Stress (GPa)
As-depo AZO film	34.32	9821	0.384	2.6098	21.2	0.52	−1.185
CBD-1 h	34.36	21542	0.272	2.6085	30.2	0.52	−0.95
CBD-2 h	34.39	37403	0.258	2.6077	31.8	0.52	−0.82
CBD-3 h	34.41	40043	0.251	2.6051	32.7	0.52	−0.37
CBD-4 h	34.42	58367	0.242	2.6021	34.1	0.52	0.15
CBD-5 h	34.45	81245	0.208	2.6003	39.5	0.52	0.47

## Data Availability

Not applicable.

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
