# Peer review of "Fabrication of Well-Aligned ZnO Nanorods with Different Reaction Times by Chemical Bath Deposition Method Applying for Photocatalysis Application"

_molecules, 2023, doi:10.3390/molecules28010397_

Round 1
Reviewer 1 Report
The paper titled ‘Fabrication of Well-aligned ZnO Nanorods with Different Reaction Times by Chemical Bath Deposition Method Applying for Photocatalysis Application’ deals with the effects of growth reaction time on the structural, optical, and photocatalytic properties of zinc oxide nanorods synthesized by a chemical bath deposition method. The materials were characterized by state-of-the-art methods and the study points out the advanced application of such materials in the environmental field. The manuscript was well organized, but several points should be taken into consideration before the publication of this article:
1. Authors should briefly discuss the reason why they chose methyl red for the photocatalytic properties examination of the synthesized materials in the Introduction part.
2. Correct formula of Zn (NO3)2·6H2O – remove the space throughout the whole manuscript
3. The authors should provide more details about the sample characterization, e.g. magnification during SEM analysis, scan rate (XRPD), etc.
4. Correct the axis labels in Figure 1.
5. Provide Rietveld analysis and comment on the parameters (perhaps the Table and corresponding discussion would be appropriate).
6. Eq. line 152 missing ‘-’ in NO3-
Author Response
Response to Reviewer Comments
We sincerely thank for your time to consider and gave good suggestions and recommendations to our manuscripts.
According to the reviewer’s comments, we revised our manuscript as follows:
Reviewer 1
Point 1:
Authors should briefly discuss the reason why they chose methyl red for the photocatalytic properties’ examination of the synthesized materials in the Introduction part.
Response 1:
In this research, the obtained ZnO nanorods were tested as photocatalyst for degradation of MR as the standard of azo dye pollutant. To clarify this purpose,
- We added the sentence of “Among the various dyes, the methyl red (MR) is one of the Azo dyes, which have been widely used in water purification, carbon dioxide reducing, textiles and some food products [5,6]. However, MR had been classified as harmful and toxic dyes because it causes the eye irradiation, skin and digestive tract if swallowed or inhaled [7,8]. Thus, it is a great concern to find out effective semiconductor metal oxides to degrade the MR efficiently from the environmental problems.” from line 31 to 36.
- Reference number [5], [6], [7], and [8] were added.
Point 2:
Correct formula of Zn(NO3)2·6H2O – remove the space throughout the whole manuscript.
Response 2:
According to the suggestion, we removed the space and revised the correct writing of Zn (NO3)2·6H2O.
Point 3:
The authors should provide more details about the sample characterization, e.g. magnification during SEM analysis, scan rate (XRPD), etc.
Response 3:
According to the suggestion, we revised as follows,
- The sentence of “The morphologies of as-deposited AZO film and ZnO nanorods were evaluated by field emission scanning electron microscopy (FE-SEM, SU-8020, Hitachi)” was revised as “The morphologies of as-deposited AZO film and ZnO nanorods were evaluated by field emission scanning electron microscopy (FE-SEM, SU-8020, Hitachi, performed in 5kV with the magnification from 20K to 100K).” from line 117 to 120.
- “The structural properties of as-deposited AZO film and ZnO nanorods were investigated by grazing incidence X-ray diffraction (GIXRD, ATX-G, Rigaku)” was revised as “The structural properties of as-deposited AZO film and ZnO nanorods were investigated by grazing incidence X-ray diffraction (GIXRD, ATX-G, Rigaku, performed in 2θ/ω sweeping from 20 to 40 degree in 2θ with a scan speed at 0.5°/min, employing a CuKα tube (λ=0.154178 nm) radiation (50 kV, 300 mA))” from line 120 to 123.
Point 4:
Correct the axis labels in Figure 1.
Response 4:
According to the suggestion, we revised the labels in Figure 1.
Point 5:
Provide Rietveld analysis and comment on the parameters (perhaps the Table and corresponding discussion would be appropriate).
Response 5:
According to the reviewer suggestion, the calculated XRD parameters were provided in Table 2 as details.
- Table 2 was added in line 145.
- The XRD parameters discussing was added, the XRD results explanation was re-written from line 130 to 158.
Point 6:
Eq. line 152 missing ‘-’ in NO3-
Response 6: According to the suggestion, we corrected the NO3 to NO3-.
Reviewer 2 Report
The manuscript of Su Wai and Li reported the photocatalytic dye degradation performance of ZnO nanorods grown on AZO film. Even though the article lacks sufficient novelty, the authors have done decent analyses and discussion. Hence, it could be recommended for publication after some revisions.
Comments:
1. Please state the importance of using Al-doped ZnO film as seed for the growth of ZnO nanorods. Undoped ZnO films could be used also as seeds as stated in this reference (J. Mater. Chem. C, 2017,5, 8059-807). The same reference gives a more detailed description on the growth mechanism of ZnO nanorods in aqueous solution. Please integrate the concept to strengthen your discussion.
2. The increase in the absorption of light by 1D ZnO nanomaterials has been reported already by several groups (Mater. Adv., 2020,1, 1253-1261; Appl. Surf. Sci., 2019, 467–468, 932—939). Please add some citations on the last sentence of the 2nd paragraph of the introduction.
3. State the specific photocatalytic reaction you did in the abstract.
4. Can you explain the diffraction shift towards higher 2theta for the 002 peaks of CBD samples?
5. Did you check if the MR dye can be adsorbed on the ZnO nanorods before photocatalysis?
Author Response
Response to Reviewer Comments
We sincerely thank for your time to consider and gave good suggestions and recommendations to our manuscripts.
According to the reviewer’s comments, we revised our manuscript as follows:
Reviewer 2:
Point 1:
Please state the importance of using Al-doped ZnO film as seed for the growth of ZnO nanorods. Undoped ZnO films could be used also as seeds as stated in this reference (J. Mater. Chem. C, 2017,5, 8059-807). The same reference gives a more detailed description on the growth mechanism of ZnO nanorods in aqueous solution. Please integrate the concept to strengthen your discussion.
Response 1:
According to the suggestion, we revised as follows:
- In introduction part, in order to clarify the purpose of using AZO seed layer, the sentence of “By means of investigating the different transparent conductive oxide substrates, the aluminum-doped zinc oxide (AZO) film has high potential for controlling the growth direction of ZnO nanorods” was revised as the sentence of “Comparing to the ZnO film, the aluminum-doped zinc oxide (AZO) film has much high thermal and chemical stability Therefore, AZO film with good crystallinity will be better seeds layer for controlling the growth direction of ZnO nanorods in the chemical bath deposition method” from line 75 to 78.
Point 2:
The increase in the absorption of light by 1D ZnO nanomaterials has been reported already by several groups (Mater. Adv., 2020,1, 1253-1261; Appl. Surf. Sci., 2019, 467–468, 932—939). Please add some citations on the last sentence of the 2nd paragraph of the introduction.
Response 2:
According to the suggestion,
- The sentence of “Pelicano, et. al was reported that modified core-shell heterostructure of ZnO nanorod could be enhanced the charge transporting capabilities with the light absorber layer and controlled the pH level to modify the different kind of ZnO nanostructures for flexible and highly efficient photovoltaic devices [40,41].” was added in line 65 to 68.
- Reference numbers of [40] and [41] were added.
Point 3:
State the specific photocatalytic reaction you did in the abstract.
Response 3:
According to the suggestion, the sentence of “The obtained zinc oxide nanorods grown for 5 hours showed the highest photocatalytic efficiency and a high first-order degradation rate of 0.0051 min-1” was revised as “Zinc oxide nanorods grown for 5 hours showed the highest degradation efficiency of methyl red under ultraviolet light and had high first-order degradation rate of 0.0051 min-1” from line 21 to 23.
Point 4:
Can you explain the diffraction shift towards higher 2theta for the 002 peaks of CBD samples?
Response 4:
In this research, the XRD diffraction shift towards the higher angle after the CBD growth.
According to the suggestion,
- The sentence “Based on biaxial strain model [47], the compressive stress (σ) in AZO film and ZnO nanorods grown with different reaction times could be expressed as
σ = × (1)
where c is the lattice constant, Cij is the elastic modulus of bulk ZnO films, and C11 = 208.8 GPa, C12 = 119.7 GPa, C13 = 104.2 GPa and C33 = 213.8 GPa. The lattice constant c also could be calculated based on residential stress and XRD patterns. The peak position of the (002) diffraction peak of as-deposited AZO film was 34.32° and it was gradually shifted from 34.36°, 34.39°, 34.41°, 34.42° to 34.45°, respectively, corresponding to ZnO nanorods with different growth reaction time. According to the Eq. (1), the calculated compressive stresses were -0.95 GPa, -0.82 GPa, - 0.37 GPa, 0.15 GPa, and 0.47 GPa, corresponding to ZnO nanorods grown from 1 hour to 5 hours. It was indicated that the compressive stress was gradually relieved in ZnO nanorods grown by the CBD process” was added from line 147 to 158.
- Reference number [47] was added.
- Equation (1) was added in line 149.
- All equation numbers were revised in sequence.
Point 5:
Did you check if the MR dye can be adsorbed on the ZnO nanorods before photocatalysis?
Response 5:
According to your suggestion, we added one additional experiment to compare the photodegradation efficiency on ZnO nanorods without UV light. To clarify the experiment,
- Photodegradation process of ZnO nanorods without UV light was added from line 110-114.
- The figure for absorption spectra of MR solution of the ZnO nanorods without UV light was added as Fig. 5 in line 236.
- The explanation for Fig. 5 was added from the line 238 to 245.
- The previous Fig.5 was changed to Fig.6 in the line 247.
- The previous Fig. 6 was changed to Fig. 7 in the line 292.
- The caption of Fig.7 was revised.
Reviewer 3 Report
This manuscript reports the preparation of the ZnO nanorod photocatalyst through the chemical bath deposition strategies. The authors investigate the effects of the grown time on structure and photocatalytic activity of the prepared ZnO nanorod photocatalysts. However, several uncertainties still need to be solved before its further consideration. Detailed comments are as following:
1. The formation rates of active radicals were strongly associated with band gaps of semiconductors, position of band edges, and the recombination rate of photogenerated electron–hole pairs. Please give more comments about the produce of active radicals for the band gaps and position of band edges.
2. The adsorption of MR by catalysts is an important part of the photocatalytic degradation of MR, thus affecting photocatalytic activity directly. The enhancement of adsorption and activation of MR by catalysts has attracted research attention. More comments about this point should be given.
3. More directly evidences using the comparison between the zinc oxide crystallines and the other ZnO should be added to show the essential roles of crystalline zinc oxide.
4. The enhancement of active sites by photocatalysts has attracted research attention. More comments about this point should be given. Moreover, it has reported to design the catalysts with high active sites, and thus enhancing the catalytic activity, for example, Molecules., 2021, 26, 2269,” Fabrication of Well-aligned ZnO Nanorods with Different Re- 2 action Times by Chemical Bath Deposition Method Applying 3 for Photocatalysis Application” J. Mater. Sci. Technol., 2020, 56, 69. The related content need to be examined to enhance the advances of this manuscript.The related content need to be examined to enhance the advances of this manuscript.
Author Response
Response to Reviewer Comments
We sincerely thank for your time to consider and gave good suggestions and recommendations to our manuscripts.
According to the reviewer’s comments, we revised our manuscript as follows:
Reviewer 3:
Point 1:
The formation rates of active radicals were strongly associated with band gaps of semiconductors, position of band edges, and the recombination rate of photogenerated electron–hole pairs. Please give more comments about the produce of active radicals for the band gaps and position of band edges.
Response 1:
According to the suggestion,
- The sentence of “Based on the Tauc’s plot equation [48], the calculated bandgap energy (Eg) values of obtained ZnO nanorods were decreased from 3.31 eV to 3.21 eV when the growth reaction time was increased from 1 hour to 5 hours. As a result, the electrons from the valence band will be much easier transferred to the conduction band of the ZnO nanorods grown with 1 hour to 5 hours, resulting in increasing the formation rate of active radicals during the photodegradation process.” was added from line 225 to line 230.
- Reference number [48] was added.
Point 2
The adsorption of MR by catalysts is an important part of the photocatalytic degradation of MR, thus affecting photocatalytic activity directly. The enhancement of adsorption and activation of MR by catalysts has attracted research attention. More comments about this point should be given.
Response 2:
According to the suggestion, we revised as follows,
- In order to understand the absorption function of MR by the ZnO nanorods, the sentence of “In order to improve the photocatalytic efficiency of ZnO, the generation rate of ROS radicals was dependent on the active surfaces of ZnO. In the wurtzite structure of ZnO, owing to the (0001) facets have high surface energy and create the more active surfaces of ZnO with the plentiful active sites, as shown in the Fig.7 (b). The ZnO polar surface of (0001) can easily absorbed OH- ions due to their surface positive charge, then reacted with the photogenerated hole (h+) to generate the ROS species, which can enhance the photocatalytic efficiency.” was added in line 308 to 314.
Point 3:
More directly evidences using the comparison between the zinc oxide crystallines and the other ZnO should be added to show the essential roles of crystalline zinc oxide.
Response 3:
According to the suggestion,
- The sentence of “According to the XRD results, the (0001) crystal facets of ZnO nanorods were enhanced with the increasing of growth reaction time. The strongest (0001) crystal facets was obtained from the ZnO nanorods grown with 5 hours.” was added from the line 314 to 316.
Point 4:
The enhancement of active sites by photocatalysts has attracted research attention. More comments about this point should be given. Moreover, it has reported to design the catalysts with high active sites, and thus enhancing the catalytic activity, for example, Molecules., 2021, 26, 2269,” Fabrication of Well-aligned ZnO Nanorods with Different Re- 2 action Times by Chemical Bath Deposition Method Applying 3 for Photocatalysis Application” J. Mater. Sci. Technol., 2020, 56, 69. The related content need to be examined to enhance the advances of this manuscript.
Response 4:
According to the reviewer suggestion,
- The sentence of “Li, et. al was reported that the modification of surface morphology and controlling the balance between specific surface area and structural defects are mainly involved in the role of photocatalytic efficiency [39]” was added from the line 60 to 63.
- The reference number of [39] was added.
- The sentence of “On the other hand, the surface area of ZnO nanorods was also increased with increasing the growth reaction times, which may have more active sites on the surface contributing to increase the degradation reaction rate” was revised to the sentence of “In addition, the volume including length and diameter of the obtained ZnO nanorods were increased when the growth reaction time was increased from 1 hour to 5 hours, which meant that there were much more active sites on the surface of the ZnO nanorods. The increase in the number of active sites could be enhanced the surface absorption ability of MR solution by absorbing more incident photons to generate the photogenerated charge carriers, thus improved the degradation reaction rate.” was added from the line 326 to 331.
Round 2
Reviewer 3 Report
The authors responsed the reviewer's questions perfectly, and the present form can be accepted.